# Machine Learning Approach for Application-Tailored Nanolubricants’ Design

**DOI:** 10.3390/nano12101765

**Published:** 2022-05-22

**Authors:** Jarosław Kałużny, Aleksandra Świetlicka, Łukasz Wojciechowski, Sławomir Boncel, Grzegorz Kinal, Tomasz Runka, Marek Nowicki, Oleksandr Stepanenko, Bartosz Gapiński, Joanna Leśniewicz, Paulina Błaszkiewicz, Krzysztof Kempa

**Affiliations:** 1Institute of Combustion Engines and Powertrains, Poznan University of Technology, 60-965 Poznań, Poland; oleksandr.stepanenko@put.poznan.pl; 2Institute of Automatic Control and Robotics, Poznan University of Technology, 60-965 Poznań, Poland; aleksandra.swietlicka@put.poznan.pl; 3Institute of Machines and Motor Vehicles, Poznan University of Technology, 60-965 Poznań, Poland; lukasz.wojciechowski@put.poznan.pl (Ł.W.); grzegorz.kinal@put.poznan.pl (G.K.); 4Department of Organic Chemistry, Bioorganic Chemistry and Biotechnology, Silesian University of Technology, 44-100 Gliwice, Poland; slawomir.boncel@polsl.pl; 5Institute of Materials Research and Quantum Engineering, Poznan University of Technology, 60-965 Poznań, Poland; tomasz.runka@put.poznan.pl; 6Institute of Physics, Poznan University of Technology, 60-965 Poznań, Poland; marek.nowicki@put.poznan.pl (M.N.); paulina.blaszkiewicz@put.poznan.pl (P.B.); 7Institute of Mechanical Technology, Poznan University of Technology, 60-965 Poznań, Poland; bartosz.gapinski@put.poznan.pl; 8Łukasiewicz Research Network—Poznan Institute of Technology, 60-654 Poznań, Poland; joanna.lesniewicz@inop.poznan.pl; 9Department of Physics Faculty, Boston College, Boston, MA 02467, USA; kempa@bc.edu

**Keywords:** carbon nanotubes, nanolubricants, machine learning

## Abstract

The fascinating tribological phenomenon of carbon nanotubes (CNTs) observed at the nanoscale was confirmed in our numerous macroscale experiments. We designed and employed CNT-containing nanolubricants strictly for polymer lubrication. In this paper, we present the experiment characterising how the CNT structure determines its lubricity on various types of polymers. There is a complex correlation between the microscopic and spectral properties of CNTs and the tribological parameters of the resulting lubricants. This confirms indirectly that the nature of the tribological mechanisms driven by the variety of CNT–polymer interactions might be far more complex than ever described before. We propose plasmonic interactions as an extension for existing models describing the tribological roles of nanomaterials. In the absence of quantitative microscopic calculations of tribological parameters, phenomenological strategies must be employed. One of the most powerful emerging numerical methods is machine learning (ML). Here, we propose to use this technique, in combination with molecular and supramolecular recognition, to understand the morphology and macro-assembly processing strategies for the targeted design of superlubricants.

## 1. Introduction

The element carbon displays exceptional diversity in atomic 3D-arrangement-driven nanoarchitectures (Figure 1) [1,2]. Structures that rely on C-sp${}^{2+e}$ hybridisation, where e=0(graphene)…1(diamond), are frequently controllable and, thus, transferable to macroscale materials possessing a one-of-a-kind combination of superb physicochemical properties [3,4]. One of the most intriguing phenomena that has been observed for carbon nanoparticles as the tribo-active components (and their formulations) at the interface of various tribo-pairs is the vanishing/ultra-low (typically less than 0.01) coefficient of friction called superlubricity [5,6,7]. Here, (i) sp${}^{2}$-carbons, that is layer-structure carbons with the most eminent graphite and its structural component graphene, (ii) sp${}^{2+e}$ diamond-like, onion-like, and fullerene-like carbons, carbon nanotubes (CNTs), and (iii) sp${}^{3}$-carbons, that is nanocrystalline diamonds, served as the solid lubricants [8,9,10]. Importantly, numerous carbon nanoparticles can be amalgamated with functional liquids to obtain novel nanofluids, that is stable nanodispersions of synergetic, thus exceptional, characteristics upon friction [11,12,13,14,15]. As for graphene, graphite, and CNTs themselves, superlubricity is attributed to the formation of atomically incommensurate contact between (variously separated) carbon nanolayers attached to the tribo-pairs. Hence, the lateral force “felt” by the atoms in the interface during sliding can be eradicated by the corresponding atom. The key disadvantage is that the structural (solid phase) superlubricity of carbon nanoparticles strongly depends on the surface perfection/crystallinity and on the given environment(s), such as surface defectiveness, contamination, temperature, humidity, and the velocity of the tribo-pairs [16]. Furthermore, this phenomenon has so far been observed predominantly at the nanoscales and microscales, while its translation into everyday life applications is yet to be achieved [7,17].

In a related effect, the ultra-low friction of water in carbon nanochannels has been observed [18,19,20,21], and the concept of fluctuation-induced quantum friction was proposed in this context [22]. This showed that friction results not only from the static roughness of the solid surface, but also from the coupling of molecular water fluctuations to electronic excitations within the solid. This mechanism might also contribute to the superfluidity effect. A solid-state analogue of such molecular to electronic fluctuations’ coupling occurs at the surface of a topological insulator and leads to the formation of a plasmon–polaron collective mode, a hybrid of the phonon and plasmon excitations [23].

The tribological mechanisms induced by carbon nanomaterials (CNMs) can be directly observed in detail, mostly only under the idealised conditions of atomic and microscale experiments. Scaling up these experiments to the friction components used in machines produces new effects, which completely change the friction conditions, an example of which is the non-controlled geometrical position of strongly anisotropic particles such as CNTs or other CNMs [6,10]. The critical impact of the CNMs’ geometrical position in relation to the surface sliding velocity can further be used to produce controlled-friction nanofluids—instead of common lubricants. Fluids allowing for a controlled switch from a low to high coefficient of friction (COF) could bring innovative elements to the machine design [24].

In our macroscale experiments, we showed that CNTs combined with liquids can considerably decrease friction [25,26,27,28,29]. This takes place either by lubricating the nanofluid films or forming interface molecular layers, both of which yield superlubricity. In particular, CNTs have emerged as powerful tribo-active nanomaterials that have high-aspect-ratio coaxial cylindrical graphene walls—hence, they are atomically smooth. Indeed, theoretical computations predict that perfectly crystalline multiwalled CNTs (MWCNTs) could become the “smoothest cylinder-in-cylinder bearings”. This behaviour was confirmed at the nanoscale by suppressing the stick–slip motion for double-walled CNTs [30,31]. In that experiment, the walls were in incommensurate contact. Nevertheless, no clear relationship can yet be found between carbon nanoparticles at the nanoscale to microscale morphology levels and the surface physicochemistry with the tribological parameters observed for macroscale polymer–metal/polymer–polymer surfaces. In order to ensure the real-life performance of this rising new class of nanolubricants, they need to be taken from molecular design to the macroscale. However, to date, most tribological tests have not been conducted via the properties-by-design approach.

Our experimental results indicate the general complexity of the tribological processes induced by the presence of CNTs in lubricants. We believe that, despite purely mechanical actions such as surface coverage (graphene), CNTs may also produce the following effects: (i) transfer of energy revealed in single-asperity contacts, in particular tribo-electrons and heat; (ii) tribochemical impact, for example interactions with common oil additives such as anti-wear additives, which lead to the activation of zinc dithiophosphate (ZDDP) [28,29]; (iii) preventing lubricant molecules from being arranged in symmetrical superstructures, thus reducing stick–slip vibrations according to the theory of shear-induced vibrations [27]; and (iv) in situ modifications of CNMs in the friction process, for example CNTs due to mechanical stress—and hence, graphene production. It is worth noting all of the abovementioned effects, and we strongly believe that there are other specific carbon-based nanolubricant effects not yet found, which can occur simultaneously, even if only trace amounts of CNMs have been introduced into the lubricant [27]. It is therefore clear that while studying the details of the tribological roles of nanoparticles is extremely important, we must simultaneously accept that quantitative calculations of friction for machine parts will not be possible in a predictable time. Moreover, contrary to the classical lubricants such as greases consisting of various oils and thickeners (whose applicability in defined friction conditions is part of the general knowledge on tribology and was studied in detail [32,33,34,35,36]), the application of CNMs still delivers unpredictable and frequently contradictory experimental results [37,38,39,40,41]. Therefore, there is an urgent need to test a new approach to the application-tailored design of nanolubricants that is based on machine learning algorithms (MLAs) [42,43].

In this paper, we propose the structure of a numerical model that includes MLAs to predict the COF for a sliding tribo-pair lubricated with defined virtual nanolubricants under defined operating conditions. Comparing different COFs, predicted via ML for a variety of virtual nanolubricants, allows the potentially most promising ones to be chosen. Specifically, we propose an MLA that evaluates the real experiment data obtained in our numerous macroscale friction experiments, that is those conducted in a tribometer, roller bearings, and fired engines where the substantial tribological impact of CNTs resulted in reduced friction, wear, and vibrations. We tested CNTs introduced into conventional grease (labelled CNT-enriched grease) or used solely as a liquid thickener (CNT-based grease).

Our ultimate goal was to come up with a numerical engineering tool to use for an application-tailored design for nanolubricants. Importantly, this approach seems to be the most complex among the currently existing artificial intelligence applications for scanning electron microscopy (SEM) image processing [44] and can be regarded as modules included in our model.

## 2. Characterisation of Materials—Structure of the Input Data for Machine Learning

The different potential of CNM friction-related applications generally grows from a twofold approach. This is based on the dispersion of certain amounts of CNMs in oils and other liquids when they are added to conventional greases (labelled CNT-enriched grease) or when the thickening ability of CNTs is used to produce grease from oils or other liquids (CNT-based grease). Nevertheless, we address the manufacture of solid CNM layers and CNM composites, including polymers or sintered metal composites as well. These applications have been experimentally tested, and some are reported in this paper in detail. Figure 2 shows the general idea behind these experiments, while Table 1 presents the CNTs that were selected for the experiments. The justification for selecting these CNTs was the need to analyse a large number of types, produced by various manufacturers. Thanks to this, we were able to test a wide range of CNT morphologies revealing the variety of tribologically relevant interactions. It is worth noting that our experiment could easily be extended by, for example, adding carbon allotropes other than CNTs, adding decorated or intentionally functionalised CNTs, or even more complex hierarchical CNM-based structures. The functional tests of nanolubricants are described and discussed in Section 3.

In general, a high BET surface area calculated from the controlled isotherm gas adsorption on the tested material surface according to the Brunauer–Emmett–Teller theory is intuitively characteristic for small-diameter CNTs, especially single-walled CNTs (SWCNTs), and corresponds to the high material reactivity. Therefore, the BET surface area may be one of the essential parameters needed to predict nanolubricant properties. Although, it is not the only decisive factor, for example sample A with MWCNTs revealing a specific surface area of ca. 304 m${}^{2}$/g acted as an oil thickener incomparably more efficiently than CNT batch J, which contained SW/DWCNTs and had a significantly higher BET surface area of 340 m${}^{2}$/g (Table 1). It is worth noting that the BET values measured for two batches of the same type of CNTs might vary significantly (compare samples B and C in Table 1), whereby no significant differences in their tribological behaviour were observed.

### 2.1. SEM Microscopy

CNTs are easily observable under high-resolution SEM magnified at 100,000× and higher. CNT images allow for a preliminary assessment of CNT purity and finding any defects of the crystal lattice (visible as elbows) and the art of spaghetti-like entanglement. We intentionally show micrographs obtained at 2000×, 10,000×, and 100,000× magnification, revealing fundamental differences in the form(s) of the agglomerates, which are usually neglected in the classical approach to studying the formation of nanolubricants (see Figure 3). Thin and long SWCNTs tend to form aligned fibrous-like architectures. On the contrary, thick MWCNTs produce spatially disordered, entangled networks [40,41,45]. Assuming that the formation of CNT agglomerates results from the physicochemical properties, especially from the surface energy of each type of nanotube, we hypothesised that the ML process would be able to find a non-trivial, hidden correlation between the form of the agglomerates and the friction-related properties.

### 2.2. Raman Spectra

In the case of CNTs, the Raman spectra are characterised by several spectral features attributed to various vibration modes [46,47,48]. We start with a low-wavenumber range of the Raman spectrum (below 350 cm${}^{-1}$), the radial breathing mode (RBM) is observed for SWCNTs). For all types of CNTs, graphene, and even amorphous carbon (soot), the intensity ratio ID/IG can be calculated (Figure 4). This value allows the assessment of CNM purity. The ID/IG ratio is an indicator of the graphitic perfection understood as the content of sp${}^{2}$-hybridised carbon atoms. A comparison of the spectroscopic features shows that the carbon materials can be divided into CNTs, graphite-like CNMs, and amorphous carbon (carbon black). In the group of CNTs, CNT J (SWCNT 40%/DWCNT 60%) is the most distinctive, for which the RBM modes are recorded, and the ID/IG ratio is the smallest (0.081). This also confirms the most perfect graphitic structure. For samples CNT I, CNT G, and CNT F, the ID/IG ratio is relatively comparable at ca. 0.54, and for sample CNT H, it is slightly higher, that is 0.724. The ID/IG ratio for the other CNTs is considerably above 1, reaching nearly 1.6 for CNT A. Moreover, regularity can be observed in the CNTs, that is longer CNTs are characterised by a lower ID/IG ratio than those observed for shorter CNTs—the exceptions are the samples CNT F and CNT G, for which the difference is negligible. Furthermore, a comparison of samples CNT A, CNT B, and CNT C shows that the purification process improves the graphitisation degree (a lower ID/IG ratio).

MLAs can test spectra for the presence of RBM peaks, the D/G value, and impurities observed at 3000 cm${}^{-1}$. In a more simplified version, only the D/G value is analysed.

## 3. Tribological Tests of CNT-Based Nanolubricants

The results of exploratory tribological tests, in some configurations, confirm the extraordinary lubricating properties of CNT-based greases. We are aware of the limitations resulting from the restrained scope of the presented results; however, the excellent tribological properties obtained for some lubricants prompted us to debate their source.

### 3.1. CNT-Based and CNT-Enriched Greases in “Block-on-Ring” Tribometers

In the first experiment (Figure 5), various CNT-based greases were prepared using high-shear mixing to disperse CNTs in CASTROL RS 10W/60 motor oil, propylene glycol (research-grade purity), and distilled water. The CNT A and CNT B nanolubricants were compared to Mobil Mobilux EP2 grease, which was used as a reference. There were 15×10×8 mm blocks machined from commercially available polymer (polyethylene, polyoxymethylene, and polyamide-6; see Figure 5 for details) plates mounted in the tribometer and pressed against a rotating steel ring (ground steel type AISI4130, Sa ≈ 0.5 μm). The dimensions of the ring counter-sample were: diameter 45 mm and width 12 mm. Pressure was applied gradually during the first seconds of the tribometer run and set to 750 N for all test configurations. The ring speed was kept constant at 115 rpm for all 30 min testing periods for each sample set, which were only lubricated with 0.1 mL of grease at the beginning of the test. The COF was measured to compare the lubricants. The 3D topography of the initial and worn surfaces were measured to estimate wear, thus both volumetric wear (volume and depth of the friction scar on the polymer block) and multi-parameter worn surface roughness analyses were conducted. The ISO 25178 height parameters, volume characteristics of the Abbott–Firestone curves, and anisotropy were used in this study of the topography. On this basis, we estimated the correlations to CNT morphology, and we tried to find parameters suitable for machine learning (ML).

In most cases, the COF value increased along with the test time. The CNT-based grease produced from motor oil thickened with purified B CNTs turned out to be a very efficient lubricant for polyamide-6 (PA6), a reliable lubricant for polyethylene (PE), and partially acceptable as a lubricant for polyoxymethylene (POM). The CNT A slurry formed in water was unsuitable for the lubrication of polymers, especially POM and PA6, that is it forced the polymers to melt, and CNTs were introduced into the polymers, producing a uniform layer on the counterpart. It is possible that, after cooling down, such a CNT–polymer composite may reveal certain desirable properties. Both pristine and purified CNTs in glycol showed an ultra-low COF in the first few minutes of the sample run; however, the COF remained stable for the next few minutes only for purified B CNTs. The pristine CNTs forced a rapid and unacceptably high increase in the friction—this behaviour might have been the effect of high Al${}_{2}$O${}_{3}$ hardness. Less probable, but still possible, this effect is the result of the unintended surface modification of CNT B with hydroxyl radicals, thus making it more compatible with the polyamide surfaces.

An additional factor important for the friction and wear of polymer–steel friction pairs was the roughness of the rubbing surfaces. An analysis of the roughness parameters of the technological surface (before the test) and of the worn surface (after the test) can be helpful in determining the level of wear and can be a starting point for the selection of the optimal characteristics to be used in ML. Figure 5 shows the surface of the selected polyamide block after the wear test.

Generally, changes in all height parameters indicate wear. The average height of the roughness parameters (Sa, Sq) increased upon testing. This means that a larger number of high peaks and deep valleys appeared on the worn surface compared to the native technological surface. This was confirmed by parameters characterising the maxima of the surface topography, that is Sp, Sv, and Sz. More detailed information comes from comparing the skewness and kurtosis. The initial surface has an Ssk of less than 0, which means that the surface is very flat, and most of the material is concentrated around the valleys. An increase in the Ssk value due to the wear test to a value above 0 (0.121) proves that the ratio of the share between the peaks and valleys of the surface has changed. The share of peaks increased, which allows us to assume that the basic wear mechanism was abrasion in the form of micro-cutting. The roughness of the peaks of the steel countersamples mainly “punched out” the core of the polymer material without significantly interfering with the valleys. The surfaces before and after the wear test are characterised by a kurtosis value above 3. Such surfaces are characterised by the presence of inordinately high peaks/deep valleys. The conclusions resulting from the analysis of Ssk and Sku coincide with the observations of the Abbott–Firestone curve characteristics. ISO 25178 volumetric parameters were used in this part of the tests. In fact, no changes in the volume parameter of the valleys were observed, which confirmed the earlier conclusion about the wear being concentrated in the core of the material. As for the material core, an increase in both parameters was observed (Vmc and Vvc). The increase in the Vvc parameter was particularly clear (approximately 75%). More free space in the material core means loss due to wear and confirms the cutting nature of the destruction.

The last part of the topographic investigation was an analysis of the surface geometric isotropy. The rose plot for the initial surface shows that the surface is characterised by a relatively low anisotropy of 42.6% (100% minus the measured isotropy). The frictional interaction made a significant change in this aspect. The worn area is already highly anisotropic, at a level of 84.6%. Summarising the topographic studies, it seems that the parameters describing the loss of the material core and the size of the peaks most appropriately describe the wear nature of the surface of the polyamide block.

In the second stage, we tested CNT-enriched greases. Regular lithium soap Mobil Mobilux EP2 grease was used as the base for a total of nine nanolubricants formulated by adding CNTs of various morphologies (CNT A…J; see Table 1). The CNT mass concentration, maintained constantly at 0.01% for this trial, was too low to reveal any relevant thickening function, although significant differences in lubricity were observed depending on the CNT morphology. The ”block-on-ring” tribometer was used for these CNT-enriched lubricity tests, in which various polymer blocks were pressed against the rotating steel ring (Figure 6). Tests were conducted using 100Cr6 steel rings with a diameter of 35 mm and a width of 9 mm, against which 15.7×10×6 mm polymer blocks were pressed. The constant load was 500 N, and the linear velocity was kept constant at 0.29 m/s (at 160 rpm). The sliding distance during each 60 min-long test was approximately 1000 m. The greases were applied only at the beginning of the test—an amount of 0.2 mL.

All CNT lubricants had lower friction coefficients than the reference lubricant when tested on PE (Figure 6). However, the results showed particularly good tribological properties of the grease with the addition of CNT A, which failed in the first trial (Figure 5), in which concentrations as high as 3 wt.% were tested. This could be explained by the CNTs and, thus, a Al${}_{2}$O${}_{3}$ concentration ten-times lower than in the preliminary test, which confirms the complexity of the CNTs’ tribological roles and justifies the use of MLAs for the efficient design of nanolubricants. For CNT A at a concentration as low as 0.01%, the COF was stabilised at an exceptionally low level of 0.01–0.015, which is significantly lower than for the other greases, including CNT B. Contrary to the PE tests, the clear impact of the CNT diameter was observed on POM, where CNT F, which had the thickest diameter in the trials of ca. 50 nm, clearly outperformed any other CNTs, as well as the reference grease.

Care should be taken when formulating unequivocal conclusions regarding the time-limited experiments presented here, although the ability of CNTs to formulate ultra-low friction nanolubricants is obvious. At this stage, we believe that ML could help find the governing correlations that would answer the question about which CNT features determine their lubrication capabilities. Again, the industrial application potential of both CNT-based and CNT-enriched greases is extremely vast due to the unique time stability of both nanolubricants.

### 3.2. CNT-Based Nanolubricants in Roller Bearings

Figure 7 shows a roller bearing test stand, with up to eight bearings mounted on a common shaft driven at a controlled speed, as well as exemplary SEM images of the tested materials.

The test performed on roller bearings lubricated by lithium grease enriched with purified CNTs where the Al${}_{2}$O${}_{3}$ catalyst was removed (sample B) delivered surprising results. Purified CNT B significantly increased wear, thus clearly promoting the formation of pockets; simultaneously, the wear of the roller bearing lubricated by grease containing pristine CNT A was very moderate. After a few hundred hours of the bearing run, grey powder was rejected from the roller bearing with CNT A and sprayed around. The Al${}_{2}$O${}_{3}$ nanospheres used for CVD CNT growth and partly remaining in the CNT A structure were separated in the friction process and converted into planar sheets (see Figure 7). Simultaneously, the ball bearing mounted on the same shaft and lubricated with the same grease showed no Al${}_{2}$O${}_{3}$ separation. We concluded that the effect of the Al${}_{2}$O${}_{3}$ nanoseed separation strongly depended on the friction conditions, in this case on the geometry of the tribological contact. However, we were not able to precisely predict the Al${}_{2}$O${}_{3}$ nanosphere’s behaviour for any possible friction condition [50]. Even more significant effects of Al${}_{2}$O${}_{3}$ were observed on the surface of the polymer, where these hard nanospheres produced adverse effects, which rapidly destroyed the polymer’s surface when supplied with grease over a certain concentration (compare Figure 5). Again, having an incomplete understanding of the co-existing tribological mechanisms could result in failures in the classical approach to the design of nanolubricants; thus, the development of ML methods becomes fully justified.

## 4. Neural Network for the Nanolubricant Design

The correlation matrix presents some obvious relations, that is that the CNT diameter and BET are strongly correlated, and this is in accordance with the theoretical calculations [51], which thus confirms the matrix’s principal usefulness (Figure 8). When analysing the oil-thickening and grease-formation abilities of CNTs, it becomes clear that, against some intuitive predictions, the CNT diameter, length, and even BET surface do not matter substantially. Therefore, we were unable to predict the lubricating performance of such a grease via similar analysis of any of the tested CNT-characterising parameters. It is thus obvious that these CNT characteristics show no significant correlation to any friction and wear parameter from the wide range of parameters that were measured. We strongly believe that MLAs will be able to discover complex multi-parameter relations, which will then allow accurate predictions to be made for the friction and wear behaviour of CNTs of a defined morphology working under defined friction conditions.

The first milestone we targeted in our project was to build a model that would allow us to predict the COF for a given nanolubricant under certain friction conditions (e.g., pressure, velocity, temperature) and for defined rubbing surfaces (e.g., material, roughness). The COF should be predicted based on a detailed characterisation of the CNTs (e.g., diameter, length, BET, and more specific test results) and on the continuous phase of the nanolubricant, which is usually the liquid in which the CNTs should be dispersed.

It is without a doubt difficult to build a single deterministic model that would consider all of the above-mentioned parameters. On the one hand, our expertise knowledge and experience allowed us to assess, via analysing certain functional relations, the quality of the material being tested; yet, on the other hand, the compiled results of individual studies turned out to be, simultaneously, completely unpredictable. It is worth noting that applying CNMs in lubrication is an emerging science; thus, a large part of the above-mentioned functional relations has not yet been discovered [27]. Hence, the idea of building a database using expertise knowledge, which will then be used to train selected MLAs with particular emphasis on artificial neural networks seems to be, undoubtedly, a promising solution. Due to the large variety of data at our disposal, that is SEM images, 3D topography of the friction track, the set of coefficients (e.g., BET or IGC parameters), and diagrams (e.g., Raman spectra or BET isotherms), we decided that it would be better to start the analyses separately for each of these types of data, particularly since, to the best of our knowledge, no such solutions have been proposed so far for this specific material in the literature.

There are ready-made solutions in the form of convolutional neural networks, such as the VGG16 [52], ResNet [53], or GoogleNet [54] networks, for input data in the form of images. Similarly, a feed-forward network can typically be intuitively constructed for a set of coefficients. It will undoubtedly be a challenge to build a neural network that will process charts such as Raman spectra or Abbott–Firestone curves, for example. Each of the data types presented in this paper can therefore be considered independently—there are similar solutions for selected types of images or sets of parameters, for example:Images of the topography of a friction track on the block have visually similar properties to maps, so trying to apply similar ML methods seems to be appropriate, that is mainly deep neural networks. Such an example solution can be found in, for example, [55].Microscopic images, such as SEM, TEM, or AFM, are used in many fields of science, but intuitively, we can immediately associate them with the field of medicine, particularly in cancer research studies. Here, solutions based on deep neural networks can also be found, e.g., [44,56,57].It seems that the easiest task would be to design an artificial neural network that would determine the COF on the basis of all kinds of parameters describing nanotubes, such as the BET area or ICG. The feed-forward neural network concept immediately comes to mind for such defined numerical inputs.Ultimately, nanotubes can be characterised via different graphs, such as a Raman spectrum or nitrogen adsorption isotherm, which we believe will be the greatest challenge for MLAs. As we have already mentioned, the most reasonable approach at this point is to use images of these graphs, as was done in, for example [58,59,60,61], or graphs generally in [62]. Another approach may be to create a feed-forward or convolutional neural network where pairs of coordinates marked on such a graph would be given as the input for this network. Perhaps the use of a convolutional neural network, where the matrix of these coordinates is treated as an image, would also be appropriate. We found no such solutions in the literature; thus, a study of such charts would be interesting.

Therefore, for individual data classes (images, charts, sets of coefficients), we were able to find independent solutions via a review of the literature and our own research experience. However, with such an extensive database, it seems that such solutions are not sufficient, as they do not consider all of the above types of data simultaneously, which distorts our main concept. Constructing a huge neural network that would process all types of data simultaneously seems to be impossible for several reasons. First, the structure of such a neural network would require enormous computing power, which may turn out to be impossible to implement. Second, if we had a huge database of, for example, images and if we wanted to show the neural network via as many patterns as possible during training, this could lead to a drastically long training time, which would not be counted in days or weeks, but possibly in months.

Therefore, we developed a general design of the ensemble model in which the artificial neural networks as described above constitute the so-called weak learners (Figure 9). Each such weak learner is thus supposed to solve a smaller problem on the way towards determining the COF, and then, in the standard approach, the meta-model can work in two ways; namely, it can be a simple average or a weighted average from the outputs of individual neural networks, or it can be an independent model that will determine the COF in an implicit way (non-deterministically). Unfortunately, in our case, the use of the average or weighted average will not necessarily work—an example is a low ID/IG calculated from the Raman spectrum, which does not necessarily provide meaningful information. A low ID/IG ratio can be obtained for CNTs with poor graphitisation, which mostly turned out to be an efficient thickener and lubricant; unfortunately, the same ID/IG ratio may characterise soot, which is entirely unsuitable for a thickening and lubricating function. On the other hand, if a low ID/IG is accompanied by a high BET, it becomes interesting, as the analysed material is not soot, but has a strongly developed surface.

As an alternative to the above ensemble model, considering all parameters of nanomaterials that are commonly recognised as important, we can imagine an extremely simplified model based only on information about the length and diameter of the nanotubes and their full Raman spectrum. These are very readily available data; therefore, such a model could be attractive for use in industrial applications while possibly being satisfactorily accurate for a relatively narrow range of the most common input parameters.

There are three types of ensemble models: bagging (built from homogeneous weak learners joined in parallel), boosting (built from homogeneous weak learners joined sequentially), and stacking (built from heterogeneous weak learners joined in parallel) [63,64]. In our case, we can interpret the type of ensemble in two ways, namely without delving into the structures of individual neural networks, we could say that the weak learners are homogeneous because they constitute the same group of classifiers; on the other hand, considering each neural network as a different structure (e.g., a convolutional network has a completely different structure to the feed-forward network), we could treat our model as being of the stacking type. The choice of meta-model is also important, apart from the selection of individual weak learners, and since weak learners are neural networks, a neural network can obviously be used as a meta-model. Other decision models, such as decision trees or gradient-boosted decision trees (XGBoost [65]), should also be considered.

## 5. Conclusions

CNMs may induce or affect numerous tribological mechanisms that take place simultaneously in the lubricating contact region and, thus, reveal complex interactions. We strongly believe that the classic approach to tribology, which relies on analytical studies and on describing the tribological roles of CNMs, should be continued even if, in a time frame that is possible to predict, the results will be qualitative rather than quantitative. In this paper, we proposed involving plasmonic interactions as one of the factors greatly affecting the friction in nanofluids. In fact, the existing insufficiencies in the knowledge on CNT tribology are one of the dominant reasons explaining the obvious gaps between outstanding CNM friction-reducing properties as observed at the atomic scale and microscale levels and the lack of large-scale industrial applications. Tribometers, bearings, and real machine tests confirmed, once again, the great potential of CNTs for reducing friction and provided widely unpredictable results that indeed depend on the nanolubricant composition and friction conditions. The COF and wear measured in the tribometer showed no significant correlation to any physicochemical CNT parameter from the wide range tested here. We thus recommend a more complex, ML-based approach to find the non-obvious multi-parameter correlations and to enable an application-tailored design of nanolubricants containing CNMs, especially CNTs.

This potential for industrial applications is promising for both CNT-based and CNT-enriched CNTs due to their unlimited time stability, which is unique among nanolubricants. Moreover, the low CNT crystallinity that turned out to be beneficial for lubricants raises the concept of large-scale and low-priced CVD process development for specific CNT growth.

## Figures and Tables

**Figure 1 nanomaterials-12-01765-f001:**
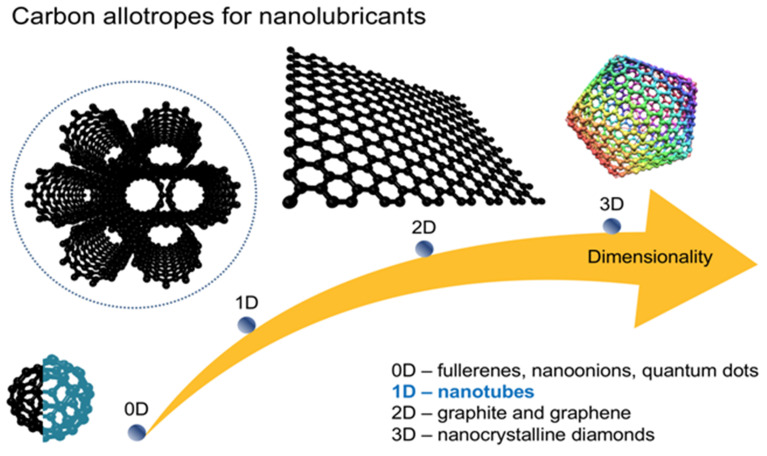
Overview of carbon allotropes that have so far been proposed as nanolubricants.

**Figure 2 nanomaterials-12-01765-f002:**
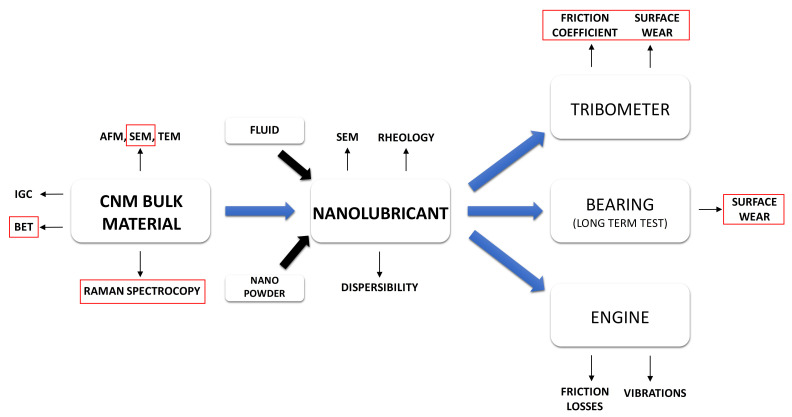
General idea of manufacturing nanolubricants and testing CNTs (only red-framed contents are discussed in this paper in detail as case-studies).

**Figure 3 nanomaterials-12-01765-f003:**
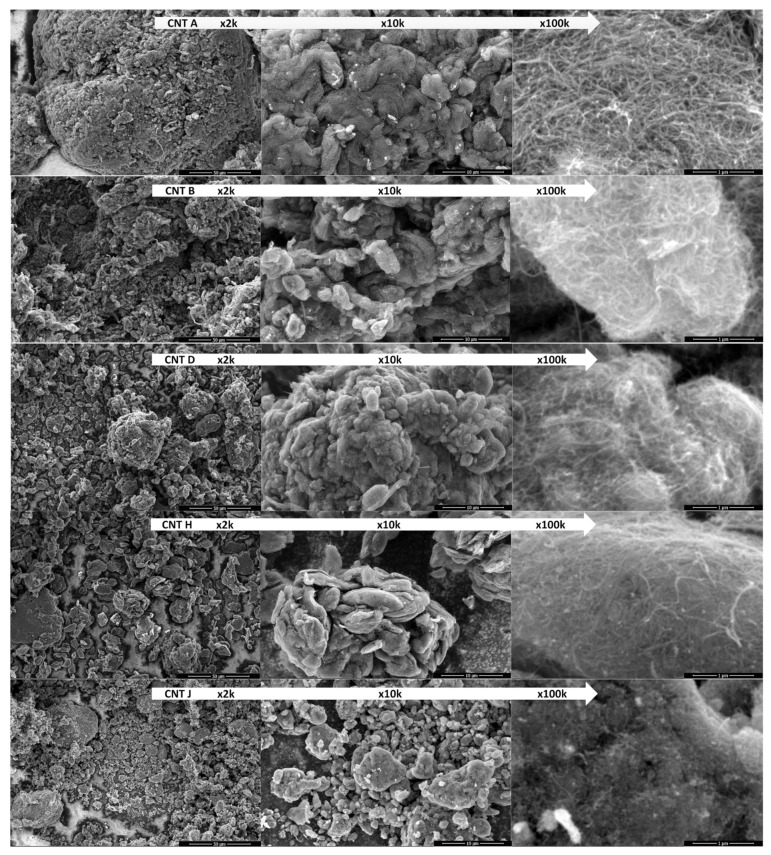
Comparison of selected CNM bulk material used for nanolubricant formulation: SEM images obtained at various magnification for the as-produced CNT powders.

**Figure 4 nanomaterials-12-01765-f004:**
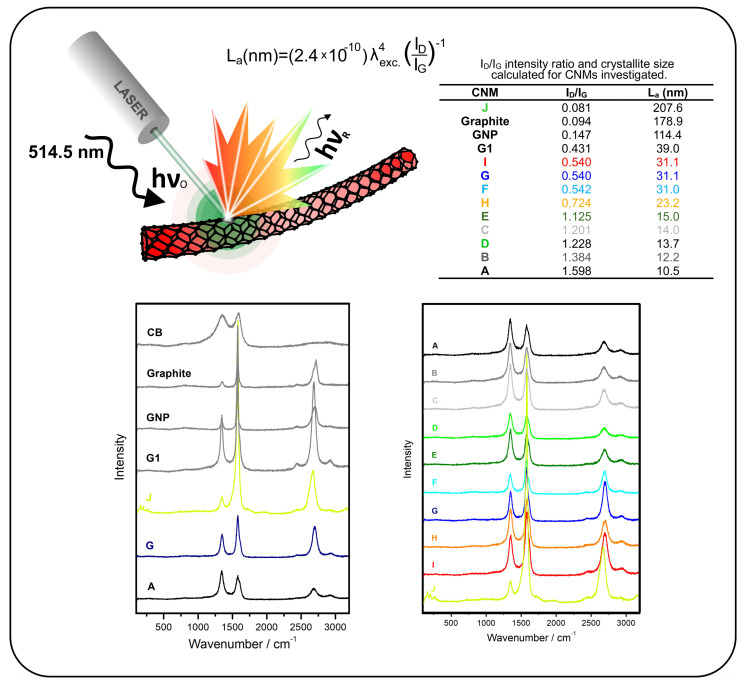
Comparison of Raman spectra for pure SW/DWCNTs and common MWCNTs of low purity, additional carbon black (CB) and graphene signals for reference; crystallite size L${}_{a}$ calculated by using the equation [49] where λ is the laser line wavelength in nanometres.

**Figure 5 nanomaterials-12-01765-f005:**
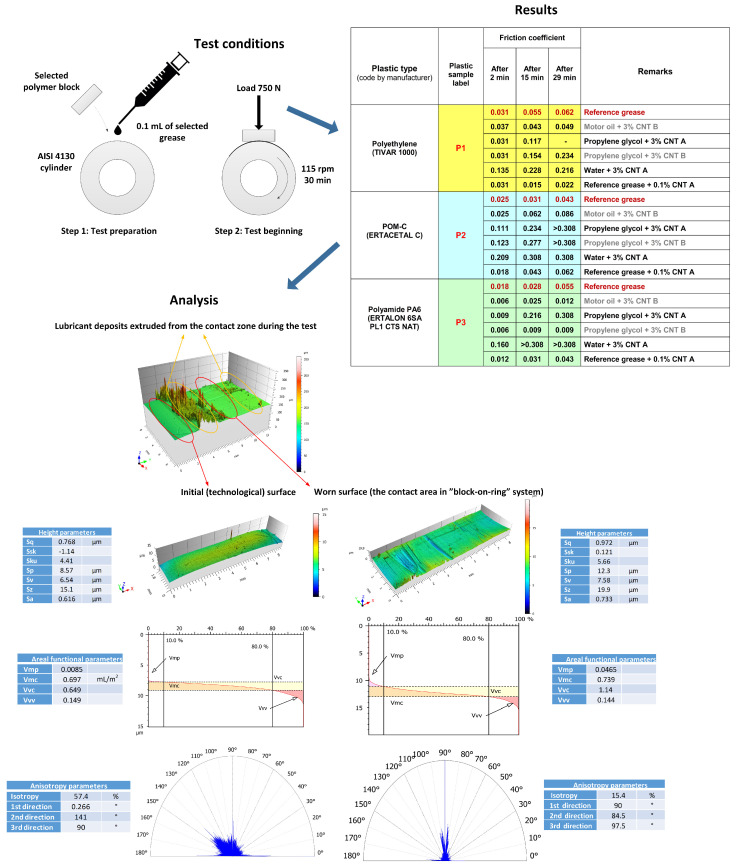
Design and results of the first “block-on-ring” tribometer experiment (roughness nomenclature: Sa (average roughness), Sq (root-mean-squared roughness), Sp (max peak height), Sv (max valley depth), Sz (max height of surface), Vmp (peak material volume), Vmc (core material volume), Vvc (core void material), Vvv (valley void volume)).

**Figure 6 nanomaterials-12-01765-f006:**
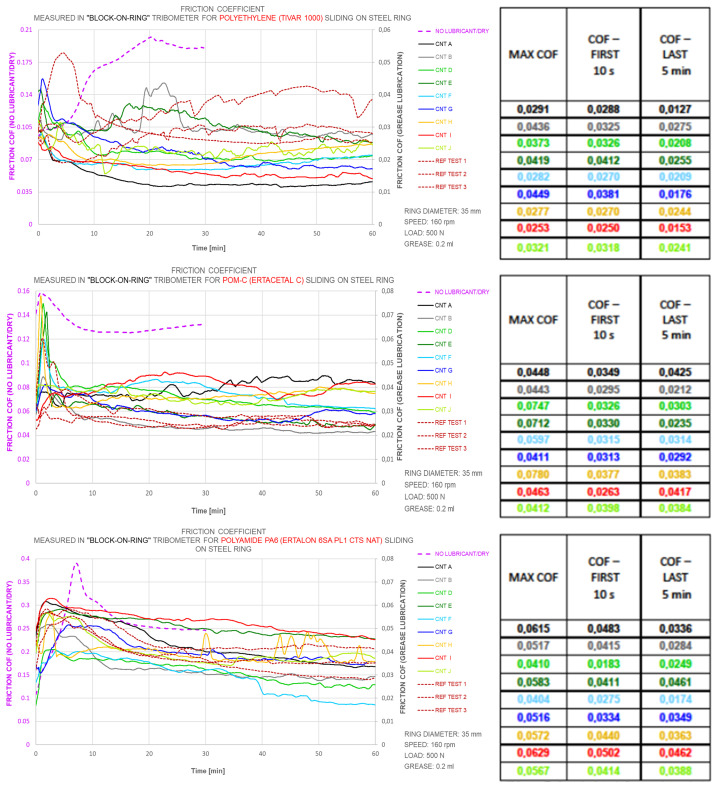
COF measured in a “block-on-ring” tribometer for various polymers sliding on a steel ring; graphs and corresponding COF values for polyethylene, polyoxymethylene, and polyamide. Comparison of CNT friction-reducing potential tests conducted for various CNT types added to the reference lithium grease in a mass concentration as low as 0.01% (COF—first 10 s and COF—last 5 min calculated as a mean value).

**Figure 7 nanomaterials-12-01765-f007:**
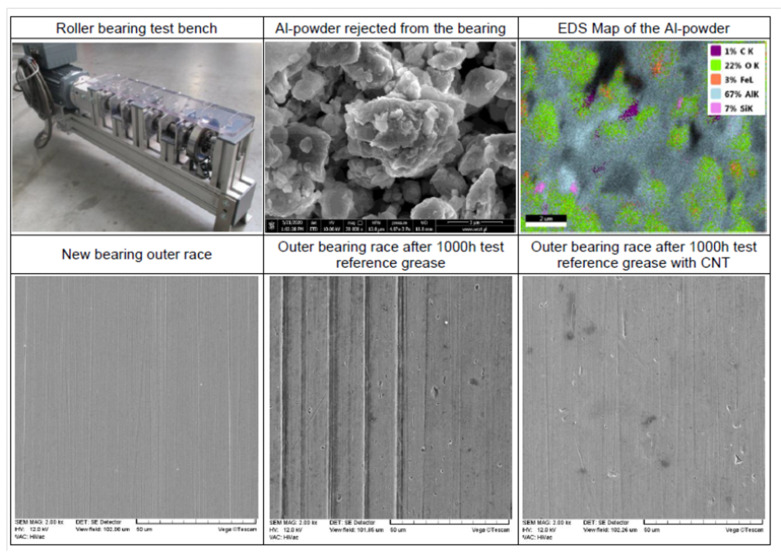
Bearing test bench and results obtained for roller bearings lubricated by reference lithium grease enriched with 0.1% of CNT A (**top row**); new ball bearing outer races and lubricated by reference lithium grease without CNT and with 0.5% CNT A after 1000 h of tests (**bottom row**).

**Figure 8 nanomaterials-12-01765-f008:**
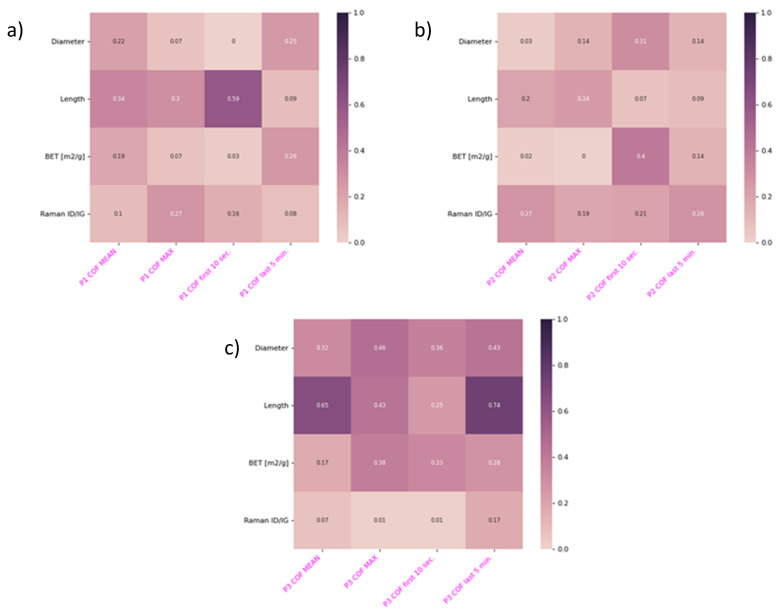
Correlation matrix for CNT diameter and length, BET surface, ID/IG ratio calculated for the CNTs’ Raman spectra, and COF reduction (tribometer) for all analysed plastics: polyethylene (**a**), POM-C (**b**), and polyamide PA6 (**c**).

**Figure 9 nanomaterials-12-01765-f009:**
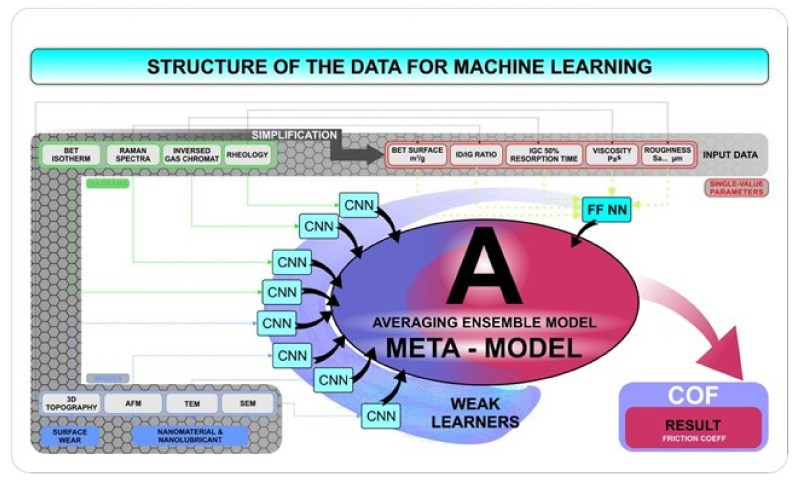
The general concept of the machine learning algorithm to predict the coefficient of friction and nanolubricant design.

**Table 1 nanomaterials-12-01765-t001:** CNT properties: length, diameter, and BET surface (measured experimentally).

Color	CNT Sample	Manufacturer	Label	Diameter * [nm]	Length * [μm]	BET [m${}^{2}$/g]
	A	Nanocyl, Sambreville, Belgium	NC 7000	9.5	1.5	300
	B	Nanocyl, Sambreville, Belgium	NC 7000 purified (Al_2_O_3_) batch 1	9.5	1.5	295
	C	Nanocyl, Sambreville, Belgium	NC 7000 purified (Al_2_O_3_) batch 2	9.5	1.5	213
	D	Cheap Tubes Inc., Grafton, VT, USA	MWCNT > 95%	<8	0.5…2.0	273
	E	Cheap Tubes Inc., Grafton, VT, USA	MWCNT > 95%	<8	10…30	261
	F	Cheap Tubes Inc., Grafton, VT, USA	MWCNT > 95%	>50	0.5…2.0	110
	G	Cheap Tubes Inc., Grafton, VT, USA	MWCNT > 95%	>50	10…20	84
	H	NanoLab, Waltham, MA, USA	PD15 L1-3	15	1…3	231
	I	NanoLab, Waltham, MA, USA	PD15 L5-20	15	5…20	220
	J	Cheap Tubes Inc., Grafton, VT, USA	SWCNT/DWCNT 60%	1–4	5…30	340

* values declared by manufacturers.

## Data Availability

Not applicable.

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
