# Peer review of "Machine Learning Approach for Application-Tailored Nanolubricants’ Design"

_nanomaterials, 2022, doi:10.3390/nano12101765_

Round 1

Reviewer 1 Report

The paper considers a very interesting and promising topic. The research is performed very well, and the presentation of the results is very good as well. The paper definitely deserves to be accepted to Nanomaterials, however, some small modifications are necessary as listed below.

1) In the Introduction, I will recommend to add some references for the first sentence, lines 16-17.

2) Also, for the sentence about the theoretical computations for CNTs, lines 60-62, I will recommend to add some references.

3) In Section 2: it would be good to provide justifications why exactly the CNTs listed in Table 2 were chosen for studies.

4) Line 119: I recommend to provide explanation what is BET.

5) In the caption of Table 1: what does the asterisk refer to?

Author Response

In this paper we present the results of our investigation and explain our interpretation. We are fully conscious that not everyone will share our view. However, we thank the reviewers for their very fair and constructive review of this manuscript. Considering the comments, some parts have been rewritten to clarify our objectives and results. Here is a detailed description of our responses to the reviewers.

The reviewers’ comments are highlighted in red.

  1. “In the Introduction, I will recommend to add some references for the first sentence, lines 16-17”

Our comment: we have added suitable references:

[1] Slepičková Kasálková, N.; Slepička, P.; Švorčík, V. Carbon nanostructures, nanolayers, and their composites. Nanomaterials 2021, 427
11. doi:10.3390/nano11092368.

[2] Hirsch, A. The era of carbon allotropes. Nature Materials 2010, 9, 868–871. doi:10.1038/nmat2885.

  1. “Also, for the sentence about the theoretical computations for CNTs, lines 60-62, I will recommend to add some references”.

Our comment: we have added suitable references:

[30] Guo,W.; Yin, J.; Qiu, H.; Guo, Y.;Wu, H.; Xue, M. Friction of low-dimensional nanomaterial systems. Friction 2014, 2, 209–225. 486
doi:10.1007/s40544-014-0064-0.

[31] Kolmogorov, A.N.; Crespi, V.H. Smoothest bearings: Interlayer sliding in multiwalled carbon nanotubes. Phys. Rev. Lett. 2000, 488
85, 4727–4730. doi:10.1103/PhysRevLett.85.4727.

  1. “In Section 2: it would be good to provide justifications why exactly the CNTs listed in Table 2 were chosen for studies”.

Our comment: The justification for selecting these CNTs was the need to analyse a large number of types, produced by various manufacturers. Thanks to this, we were able to test CNTs with a wide spectrum of parameters. Particular attention is drawn to the large differences in BET values.

  1. “Line 119: I recommend to provide explanation what is BET”.

Our comment: We rephrased the sentence where abbreviation BET appears for the first time in the article:

In general, a high BET surface area calculated from the controlled isotherm gas adsorption on the tested material surface according to the Brunauer–Emmett–Teller theory is intuitively characteristic for small-diameter CNTs, especially single-walled CNTs (SWCNTs), and corresponds to the high material reactivity.

  1. “In the caption of Table 1: what does the asterisk refer to?”

Our comment: The asterisk was supposed to refer to the columns regarding the length and diameter of the CNTs. To be clearly understood, we have changed the description of the table and have placed an explanation under it about the asterisk.

Reviewer 2 Report

Comment on “Machine learning approach for the application-tailored nanolubricants design” by Kaluzny et al.

Tribological mechanisms induced by carbon nanomaterials are interesting to the researchers for lubricants due to that fluids allowing for a controlled switch from a low to high friction coefficient could bring the innovative machine element design. In experiments, there was no clear relationship between carbon nanoparticles at the nano-to-microscale morphology level and surface physicochemical properties observed in macro-scale polymer-metal or polymer-polymer interfaces. Now Kaluzny et al. used machine learning algorithms to predict the friction coefficient for the tribo-pair sliding lubricated specific nanolubricants under different operating conditions. The machine learning technique was combined with the molecular and supra molecular recognition to understand morphology and macro-assembly processing strategies for the superlubricant design. They also found from their experiments the general complexity of the tribological processes induced by the presence of CNTs in the lubricants. The modeling methods proposed by the authors seems to be striking and inspiring. However, I have not seen any results from machine learning and thus the conclusion of the manuscript is incomplete and was not fully supported by the evidence provided in the main text. Due to these shortcomings, the manuscript could not be accepted in the present form. To achieve the standard of publication, a major revision is required.

The following concerns which I have should be addressed to improve the manuscript. They should be properly included in the revised manuscript.

(1) The three subfigures in Figure 8 can be easily confused. For clarity, please label the three subfigures using “(a)”,“(b)” and “(c)” and explain them in the figure caption.

(2) In Section 4 Neural network for the nanolubricant design, the authors described too many background knowledges but the design scheme was not stated clearly. Please use languages clearly and concisely to illustrate the neural network used for the nanolubricant design in the main text. The description from Line 300 to Line 359 could be deleted or substituted by a few of simplified and concise sentences.

(3) The title of the manuscript is “Machine learning approach for the application-tailored nanolubricants design”, however, I have not seen any promising results from the machine learning method proposed in the manuscript. It seems that the authors have presented some experimental results and proposed the readers that one can select the machine learning schemes to explain the experimental results. If so the manuscript would make no sense. Otherwise, the results of the comparisons of friction coefficients between the experiments and machine learning scheme predictions should be presented and discussed in detail.

(4) In the machine learning method, the machine needs a large amount of numerical data to “learn”. The more data used to “learn”, the more accurate the prediction. However, there are only 10 sets of data used in the manuscript. They are too few to obtain a good “learning”. I suggest the authors collect more literature data for their machine learning investigation.

(5) The authors intend to use BET data as input for the machine learning, this is very good. However, how to directly use SEM images to “train” the machine is a challenging task because the machine only accepts number and reading figure is a very difficult thing to be done by a machine. How did the authors abstract the SEM images and other figures to numerical digits for the machine?

(6) Because there were some experimental results reported in the manuscript and they are important, the experiments are better to be included in the title of the manuscript for the readers to clearly understand the content of the manuscript. Otherwise, one may regard the manuscript as a theoretical modelling paper.

(7) Some sentences are too long to be easily understood. Please rewrite the manuscript using fluent English to express the methods and results.

(8) Please make the format of the article titles in the reference list identical. For instance, the article title in Ref. 1 “Superior Thermal Conductivity of Single-Layer Graphene” should be revised to “Superior thermal conductivity of single-layer graphene”. The same format error could also be found in Refs. 5, 9, 13, 16, 17, 26, 30, 31, 33, 36, 38, 39, 40, 53, 55, and 56.

Author Response

In this paper we present the results of our investigation and explain our interpretation. We are fully conscious that not everyone will share our view. However, we thank the reviewers for their very fair and constructive review of this manuscript. Considering the comments, some parts have been rewritten to clarify our objectives and results. Here is a detailed description of our responses to the reviewers.

The reviewers’ comments are highlighted in red.

  1. “The three subfigures in Figure 8 can be easily confused. For clarity, please label the three subfigures using “(a)”,“(b)” and “(c)” and explain them in the figure caption”.

Our comment: To improve the readability of the manuscript we have changed this figure according to the suggestion.

  1. “In Section 4 Neural network for the nanolubricant design, the authors described too many background knowledges but the design scheme was not stated clearly. Please use languages clearly and concisely to illustrate the neural network used for the nanolubricant design in the main text. The description from Line 300 to Line 359 could be deleted or substituted by a few of simplified and concise sentences”.

Our comment:

Our concept for this paper was not to present one more study of the lubricating performance of CNTs in isolated test conditions, since a variety of such tests is already present in the literature. Unfortunately, neither the literature review nor our own year-long tribological studies allowed us to predict the lubricating potential of CNTs under defined friction conditions, depending on their morphology. In other words, even with all the current knowledge, nobody is able to predict which type of CNTs will be optimal for a defined tribological application. This is an obvious weakness, since we need clear guidelines for efficient engineering, as we do for the selection of the optimal lubricant viscosity depending on the load and speed or type of grease thickener chosen for the environmental conditions. We, the authors, have already accepted the fact when we start to believe that the impact of the morphology of CNTs on friction can be predicted, the next experiment can then deliver absolutely surprising results, destroying our vision and prediction power.

We strongly believe that emerging and powerful artificial intelligence tools will provide the solution, opening the way towards industrial applications for CNTs in nanolubricants. The development of machine learning tools for the design of carbon nanolubricants now constitutes a pioneering field of science. According to our best knowledge, nobody has yet described a functioning machine learning tool for the design of nanolubricants. In this manuscript, we have started the discussion about how to do this properly and efficiently, which is why our discussion stems from the basics of machine learning methods. We do not have a functioning machine learning tool for the design of carbon nanolubricants, and we do not believe that anybody does, which is why we discuss how to build one. This is the current state of knowledge. Therefore, we will not remove lines 300 to 359 and reduce the manuscript to a case study about the lubricating potential of CNTs. Unfortunately, at the moment, we also are not able to replace the text in the mentioned lines with the conclusions.

  1. The title of the manuscript is “Machine learning approach for the application-tailored nanolubricants design”, however, I have not seen any promising results from the machine learning method proposed in the manuscript. It seems that the authors have presented some experimental results and proposed the readers that one can select the machine learning schemes to explain the experimental results. If so the manuscript would make no sense. Otherwise, the results of the comparisons of friction coefficients between the experiments and machine learning scheme predictions should be presented and discussed in detail.

Our concept for the manuscript is explained in our answer to point 2.

In our opinion, the machine learning approach presented in the manuscript is very complex in comparison to the existing tools, for example, the powerful medical system helping to detect skin cancer – melanoma, which involves using machine learning for the analysis of simple smartphone pictures of the skin [https://news.mit.edu/2021/artificial-intelligence-tool-can-help-detect-melanoma-0402]. In our case, the data set for artificial intelligence assisted analysis is incomparably wider, making the model more complex, even shifting it to the borders of feasibility for the current stage of science and technology. It makes a lot of sense to discuss the structure of the algorithms creating the machine learning tool for the design of nanolubricants at the current stage of science. We find the suggestion surreal that this discussion is not a topic for this manuscript and only the final results of a functioning machine learning tool for the design of nanolubricants can be published

  1. In the machine learning method, the machine needs a large amount of numerical data to “learn”. The more data used to “learn”, the more accurate the prediction. However, there are only 10 sets of data used in the manuscript. They are too few to obtain a good “learning”. I suggest the authors collect more literature data for their machine learning investigation.

As mentioned above in the answers to points 2 and 3, our concept of the manuscript is to discuss how to design a machine learning model that is able to predict the friction coefficient of defined nanolubricants in defined friction conditions. The presented friction coefficient values measured for various polymers lubricated by various CNTs create a background and motivation for the machine learning application. Extending this data to create the learning set for the artificial intelligence model is the reasonable next stage after this model has been constituted.

  1. The authors intend to use BET data as input for the machine learning, this is very good. However, how to directly use SEM images to “train” the machine is a challenging task because the machine only accepts number and reading figure is a very difficult thing to be done by a machine. How did the authors abstract the SEM images and other figures to numerical digits for the machine?

Actually, the machine learning analysis of the SEM pictures presenting the CNT powder used in the nanolubricant formulations seems to be one of the central modules of the proposed machine learning model for the design of nanolubricants. Currently, our work focuses on this field. In general, pictures processing by machine learning is not a novelty, and in some applications outstanding results have already been obtained, as mentioned above [[https://news.mit.edu/2021/artificial-intelligence-tool-can-help-detect-melanoma-0402].

The task of the convolutional layers of artificial neural networks is to pre-process images. This convolution layer is used to detect the patterns revealed in the individual images used for training. The images are then represented as a two-dimensional (black and white images) or three-dimensional (colour images) matrix, each point of which is described by a value that specifies a colour from the RGB palette.

There are examples of applications of convolutional neural networks for processing images representing the course of certain functions, for example, in the works of M. H. Modarres et al., E. Minelli et al. or J. Ede, included in the bibliography of the article. Our task is to check whether the analysis of such complex images by convolutional neural networks is actually possible.

  1. Because there were some experimental results reported in the manuscript and they are important, the experiments are better to be included in the title of the manuscript for the readers to clearly understand the content of the manuscript. Otherwise, one may regard the manuscript as a theoretical modelling paper.

As mentioned above, we treat the manuscript as an opening for the discussion about how to build a model involving machine learning for the design of nanolubricants. This is the scientific novelty, and the original tribometer results presented only provide a background. These could be regarded as groundbreaking ten years ago, but now there are already a number of similar reports confirming the existence of the outstanding lubricity of CNTs in isolated friction conditions. Now the challenge is to convert this knowledge into large scale engineering and, with this paper, we take a step in this direction.

  1. Some sentences are too long to be easily understood. Please rewrite the manuscript using fluent English to express the methods and results.

We have checked the manuscript again and simplified the language as far as possible. Good readability of the manuscript is an essential factor; it is challenging to combine the language of tribology and machine learning in our multidisciplinary paper.

  1. Please make the format of the article titles in the reference list identical. For instance, the article title in Ref. 1 “Superior Thermal Conductivity of Single-Layer Graphene” should be revised to “Superior thermal conductivity of single-layer graphene”. The same format error could also be found in Refs. 5, 9, 13, 16, 17, 26, 30, 31, 33, 36, 38, 39, 40, 53, 55, and 56.

The proposed changes have been introduced in the revised manuscript.

Round 2

Reviewer 2 Report

The authors have revised their manuscript carefully according to the comments and suggestions from the Reviewer. They have revised the figures and figure captions for clarity and the format of the References has been corrected. The modifications have made the manuscript more significant and clearer. Now the quality of the revised manuscript is adequately good and thus I think the revised version of the manuscript could be accepted for publication in Nanomaterials.